# Geriatric Care Management System Powered by the IoT and Computer Vision Techniques

**DOI:** 10.3390/healthcare11081152

**Published:** 2023-04-17

**Authors:** Agne Paulauskaite-Taraseviciene, Julius Siaulys, Kristina Sutiene, Titas Petravicius, Skirmantas Navickas, Marius Oliandra, Andrius Rapalis, Justinas Balciunas

**Affiliations:** 1Faculty of Informatics, Kaunas University of Technology, Studentu 50, 51368 Kaunas, Lithuania; 2Department of Mathematical Modeling, Kaunas University of Technology, Studentu 50, 51368 Kaunas, Lithuania; 3Biomedical Engineering Institute, Kaunas University of Technology, K. Barsausko 59, 51423 Kaunas, Lithuania; 4Faculty of Electrical and Electronics Engineering, Kaunas University of Technology, Studentu 48, 51367 Kaunas, Lithuania; 5Faculty of Medicine, Vilnius University, Universiteto 3, 01513 Vilnius, Lithuania

**Keywords:** geriatric care, IoT, vital parameters, posture recognition, image recognition, deep learning, non-contact monitoring

## Abstract

The digitalisation of geriatric care refers to the use of emerging technologies to manage and provide person-centered care to the elderly by collecting patients’ data electronically and using them to streamline the care process, which improves the overall quality, accuracy, and efficiency of healthcare. In many countries, healthcare providers still rely on the manual measurement of bioparameters, inconsistent monitoring, and paper-based care plans to manage and deliver care to elderly patients. This can lead to a number of problems, including incomplete and inaccurate record-keeping, errors, and delays in identifying and resolving health problems. The purpose of this study is to develop a geriatric care management system that combines signals from various wearable sensors, noncontact measurement devices, and image recognition techniques to monitor and detect changes in the health status of a person. The system relies on deep learning algorithms and the Internet of Things (IoT) to identify the patient and their six most pertinent poses. In addition, the algorithm has been developed to monitor changes in the patient’s position over a longer period of time, which could be important for detecting health problems in a timely manner and taking appropriate measures. Finally, based on expert knowledge and a priori rules integrated in a decision tree-based model, the automated final decision on the status of nursing care plan is generated to support nursing staff.

## 1. Introduction

Geriatric care is a field of healthcare that focuses on the physical, mental, and social needs of older adults. As people age, they may experience physical, cognitive, and social changes that require special care and support. Geriatric care is based on the specific needs of older adults and aims to improve their health and well-being as well as manage age-related diseases and conditions so that they can maintain their independence, quality of life, and overall comfort. Such care often involves a multidisciplinary approach with care provided by a team of health care professionals, including physicians, nurses, therapists, and social workers, who are trained in gerontology and geriatrics [1,2]. The estimated number of dependent people in need of some form of long-term care in Europe is 30.8 million, and this is expected to increase to 38 million by 2050. Furthermore, the expected shortage of nurses will reach 2.3 million in 2030. By 2080, the population aged over 80 years and older in Europe will have multiplied by 2.5. It should be noted that the majority of dependent patients suffer from Alzheimer’s and chronic diseases, such as past myocardial infarction, congestive heart failure, cardiac arrhythmia, renal failure, and chronic pulmonary disease, have an increased risk of mortality in nursing homes [3].

Currently, the main problems are caused by the absence of tools to design automated care plans. The problems identified are related to the lack of digital evidence-based protocols for different situations and the nonadherence to existing protocols by nursing staff. Typically, an individualised nursing care plan is developed for the elderly patient upon admission to meet their needs. This plan is developed based on a thorough assessment of the person’s medical history and evidence-based care practices. As elderly individuals reside in nursing homes, it is common for their health to decline, which makes it crucial to monitor their health status while they are there. Thus, caregivers must regularly check important biometric data, such as blood pressure, heart rate, body temperature, and respiratory rate. Collecting and documenting patient vital signs data manually is a relatively slow and therefore inefficient process. Depending on the types of vital signs, it usually takes up to five minutes to assess three to six vital signs [4]. Moreover, this information is usually documented in paper form separately from the nursing care plans, and therefore, the whole process takes up to 13 min per patient [5]. Furthermore, care plans have to be regularly re-evaluated by comparing current and historical health records to look for abnormalities and changes that could have clinical significance. However, biometric data are documented separately from nursing care plans and records of doctors. With such fragmented data sources, the process is human-dependent, highly inefficient, and cumbersome and can take up to 37 min per patient [5,6]. Moreover, in the absence of a systematic approach in geriatric care management, it becomes challenging to quickly capture monitoring data and act on them. This can cause caregivers to miss any unusual changes in the biometric data, leading to delays in administering treatment.

During the course of our research, several hospices from Latvia, Estonia, and Poland (e.g., Orpea) were contacted, and it was concluded that geriatric care management systems with a digital care plan and remote monitoring solutions are currently not available in these markets. Facilities rely on outdated software that was developed for inpatient hospital services without taking into account the nursing care plan. In particular, in Scandinavian and UK markets (e.g., Appva), some tools have been developed that include a simple digitised nursing care plan without remote monitoring or decision support capabilities; however, none of these companies have shown an interest in providing the service in the Baltic States. Therefore, in many countries, including the Baltic States, nurses use paper-based care plan templates and manually prepare time-consuming documents. Consequently, data loss and missing information in care plans are common problems. Based on the problems identified during oral interviews and discussions with various stakeholders in Lithuania, the following needs for long-term care at home and in specialised institutions have been narrowed down as the most recurrent and yet relatively possible to complete with limited funding: (1) easily create nursing care plans for new patients with action protocols for nursing staff; (2) ensure adherence to and traceability of the execution of the protocols; (3) automate patient monitoring; (4) reduce manual paper documentation; (5) easily adapt nursing care plans according to changes in the health of the patient; and (6) enable a transition from reactive care to proactive care.

Digitalised care systems could be a solution to meet the multidimensional need to monitor whether elderly patients in geriatric care facilities are receiving optimal care, thus monitoring patients more efficiently and providing personalised care. Digitalisation also helps a relatively small number of healthcare workers to reduce the need for repetitive manual work and use the collected data for proactive decision making. Furthermore, the combination of Internet of Things (IoT) and artificial intelligence (AI) technologies can aid in the analysis of data and ensure continuous monitoring of elderly patients to positively impact their care and outcomes [7,8,9,10]. By collecting data on patient activity and health, advanced AI algorithms can analyse patterns and detect deviations from normal behaviour, allowing caregivers to respond in a timely manner.

In this study, we propose an intelligent geriatric care management system based on AI and IoT to track and detect changes in the health status of elderly patients, thus ensuring efficient digitalisation of personalised care plans. The proposed solution can be used to tackle two of the most urgent problems in the area: nursing staff shortages and the costly and inefficient long-term care process. Although home care for dependent and elderly people is becoming more and more popular, it is still not a viable option for everyone due to the expensive infrastructure required and the difficulties in gaining access to their homes in an emergency. Even if people choose to live in a nursing home, it is still difficult to monitor, care, and treat elderly residents on a regular basis. With the growing demand for healthcare nurses, fragmented remote health monitoring tools, and lack of existing solutions for real-time modifications of nursing care plans, it is crucial to have a cost-effective and semi-autonomous solution available in the market.

## 2. Related Works

In recent years, there has been a growing interest in the development of digital health solutions to support older people and promote healthy ageing [11,12]. However, elderly individuals are more likely to develop diseases such as dementia, diabetes, and cataracts, suffer from physical and cognitive impairments, and have low levels of physical activity, all of which lead to a continuous decline in their health. This makes it difficult for staff to keep track of elderly people, to monitor changes in their health, to record and store all readings systematically, and to always react quickly and appropriately to the changes and adjust the care plan. Furthermore, as life expectancy continues to increase, the need for nurses working in geriatrics is also increasing. As such, remote monitoring and wearable devices can be used to measure vital signals, evaluate physical activity, and inform caregivers or physicians about changes in their health, which aids in the early detection of health risks [13,14].

### 2.1. The Use of Wearable Devices

Recently, wearable technology has benefited from technological progress, as the size of devices has significantly reduced, while the efficiency of energy consumption has improved simultaneously [15]. In particular, wearable technology can be used for a variety of purposes, ranging from keeping track of physical activity to monitoring clinically important health and safety data. Wearable devices provide real-time monitoring of the wearer’s walking speed, respiratory rate, measuring sleep, energy expenditure, blood oxygen and pressure, and other related parameters [16]. Such devices can also be useful tools for people living with heart failure to facilitate exercise and recovery [17,18]. Comparatively, a study demonstrated the strong potential for improvement in healthcare through the use of wearable activity monitors in oncology trials [19]. The use of wearable technology to identify gait characteristics is another intriguing example [20], where lower limb joint angles and stride length were measured simultaneously with a prototype wearable sensor system. The study [21] investigated how a wearable device could help physicians to optimize antiepileptic treatment and prevent patients from sudden unexpected death due to epilepsy. For particular groups of individuals that suffer from chronic disease such as diabetes mellitus, cardiac disease, or chronic obstructive pulmonary disease, wearables may be used to monitor changes in health symptoms during treatment and may contribute to the personalisation of healthcare [22,23,24]. The use of wearables within a group of elderly population brings additional challenges. For example, it is very important to detect falls, which has already become a topic of particular importance in this field. For example, in [25], a framework was proposed for edge computing to detect individual’s falls using real-time monitoring by cost-effective wearable sensors. For this purpose, an IoT-based system that makes the use of big data, cloud computing, wireless sensor networks, and smart devices was developed and integrated with an LSTM model, showing very promising results for the detection of falls by elderly people in indoor circumstances. The validity and reliability of wearables have been addressed by many studies focusing on different classes of devices used to measure activity or biometric data [26,27,28,29]. Apparently, there is no consensus among researchers, as findings depend on the manufacturer, device type, and the purpose for which it was used. This is also true because devices are constantly being upgraded to new models, which suggests that their validity and reliability will improve with time.

### 2.2. Contactless Measurement of Vital Signs

There are still some concerns regarding the reliability and accuracy of wearables to detect physical activity and evaluate health-related outcomes within elderly individuals, as they are generally designed primarily to collect biometric information during activities of daily living in the general population [30,31,32,33]. First, the ability of older people to recognise the need for wearables and properly use them poses new challenges. Second, the high prevalence of different diseases in this population and the heterogeneity associated with their lifestyle, needs, preferences, and health point to the need for wearable devices that are valid and reliable and that can accurately measure and monitor important signals. Additionally, taking into account the problems associated with time-inefficient work in care homes, contactless monitoring of vital signs may be beneficial for healthcare [34,35,36]. In particular, contactless measurement techniques can be applied to measure the respiratory rate and monitor the heart rate variability, which is one of the fourth most important vital parameters [37]. Monitoring the respiratory rate is useful for the recognition of psychophysiological conditions, the treatment of chronic diseases of the respiratory system, and the recognition of dangerous conditions [38,39]. Combining respiratory rate and heart rate data provides even more useful information on the condition of the cardiovascular system [40,41]. The most promising method of noncontact monitoring of the respiratory process is through infrared and near-infrared cameras [42,43]. An infrared camera is a device that can capture small temperature changes on the surface of an object and/or in the environment. This device can record the temperature fluctuations of airflow from the mouth or nose. Infrared cameras can successfully measure the respiration rate if advanced computer vision algorithms that are insensitive to constantly varying lighting and temperature conditions are applied.

### 2.3. Benefits of Computer Vision Techniques

Image recognition is one of the main methods used to determine an individual’s pose and activity. The use of pose estimation technology in geriatric care offers several advantages, including the continuous monitoring of patients, early detection of potential health problems, essential data on the patient’s movements, and, in particular, the detection of extra situations (e.g., the person is lying on the ground and not moving) [44]. Pose estimation algorithms vary in complexity and accuracy, ranging from simple rule-based algorithms to more complex deep-learning-based algorithms. Simple algorithms may be faster and easier to implement but typically they are not as accurate as more complex ones. Deep-learning-based algorithms, on the other hand, may provide more accurate results but may be more computationally intensive and require large amounts of training data. Comparatively, deep-learning-based methods have shown great potential for improving the accuracy of human posture recognition, for both single individuals [45,46] and multiple individuals [47,48] in images or videos. In particular, methods such as the multisource deep model [49], the position refinement model [50,51], and the stacked hourglass network [52] have demonstrated the effectiveness of deep learning in human posture recognition. These methods use convolutional neural networks to extract features from input images and estimate the positions of human joints. However, the early detection of falls [53,54,55] is one of the most important functions of the geriatric care system as it allows prompt medical assistance to be provided and can prevent further injuries. Human fall detection systems can help to identify when a fall has occurred and alert caregivers or emergency services immediately. Therefore, various types of fall detection and prediction systems suggested in the field not only rely on image recognition techniques [42,56,57] but also employ other information sources, for example, biological factors or signals obtained by wearable devices that are more commonly used for fall risk assessments [58,59]. Although computer vision techniques have been used widely and very successfully in medicine, the monitoring and identification of patients in nursing homes should take into account the fact that image capture devices cannot always be used to track patients (e.g., hygiene rooms) according to privacy and ethical requirements [60,61]. In addition, capturing certain information with cameras may not always be possible due to changes in the environment or, for instance, in cases when the person reappears or is partially obscured by other objects, which poses the additional challenge of re-identifying the same individual. Therefore, it is important to determine which factors may be automatically recorded and tracked over time utilising image processing technology. It is also crucial that the solution is quick. As such, it is essential to carefully assess the trade-off between precision and speed in order to choose a solution that meets the specific requirements of the application.

## 3. Materials and Methods

The proposed solution includes (1) an IoT module with integrated wearable and contactless devices; (2) an AI module that utilises deep learning architectures for the image recognition of patient posture and activity; and (3) a decision support module for generating the patient-personalised nursing care plan.

An IoT module has been developed to monitor and transfer data in real time. It consists of sensors connected to an Arduino microprocessor to monitor the patient’s vital signs. This module integrates not only body-worn devices that are networked but also a number of remote devices for monitoring health data. In general, such devices can collect and transmit the collected data, such as heart rate, body temperature, and physical activity, to a remote system or application, usually through wireless connectivity (e.g., Bluetooth, Wi-Fi). Some wearable health devices also have built-in sensors and algorithms that can perform basic health assessments, such as tracking sleep patterns, counting steps taken, and estimating calorie expenditure.

In this study, four IoT devices, a Fitbit wristband, smart scale, smart blood pressure device, and a camera, were used to monitor the health of elderly patients in a nursing home (Table 1). Data collected from these devices were sent to the server and processed to obtain the final decision (Figure 1).

A patient room in a hospital for the elderly was equipped with cameras to continuously monitor the status of the patients in real time. The video footage from these cameras was sent directly to a server where it was stored, processed, and analysed using image processing algorithms. This was necessary to monitor patients’ motor activity, changes, or progress in movement and consequently make the necessary changes to the care plan or react in emergency situations such as falls, pressure sores, etc. In parallel to the cameras, the patients were also given Fitbit wristbands for the additional monitoring of physiological parameters. These wristbands were equipped with sensors to monitor the patient’s vital signs, such as heart rate and respiratory rate. The data from the Fitbit bracelets were sent to Google Cloud and then to a server using APIs. The geriatric nurse also used specialised equipment to monitor the patients’ weight and blood pressure. Withings body+ connected scales make it easy to monitor weight, BMI, fat, water, and body mass, which is later automatically synchronised with the smartphone via Wi-Fi or Bluetooth. In particular, monitoring the following parameters is important for patients at risk of complications such as high blood pressure, diabetes, etc.

One of the main limitations is that off-the-shelf IoT devices do not offer the option of sending data directly to a third-party server. As a result, all data must first pass through the provider’s cloud services and use their API. This also leads to software limitations, such as only allowing one IoT device of a certain type per account, making the data collection pipeline more complex than is necessary.

Data captured by all smart devices not only digitalise the tracking of key physiological parameters but also enable the investigation of dependencies between these indicators and a patient’s health status or its change, but only when a statistically reliable sample is collected. If computer-vision-based health monitoring is involved, real-time visual information collection must include data storage and analysis [44,62,63]. For the experiment, data collection started on 15 September 2022 and data were uploaded to the server Dell PowerEdge R7525 (AMD EPYC 7452 32-Core Processor/2350 MHz; 512 GB RAM; NVIDIA GA100 [A100 PCIe 40 GB], 2 × 450 GB SSD; 2 × 25 Gbps LAN MT27800 Family [ConnectX-5] 2 × 100 GBps [ConnectX-6]). In total, 1.412 TB of data were accumulated during the observation period between 15 September 2022 and 28 December 2022. In addition to the data collected from the IoT devices, the system also allowed manual input from healthcare personnel. This included additional parameters that were not captured by the IoT devices, such as bedsores, changes in eating habits, changes in bowel movements, etc. These data were entered into an Excel spreadsheet by healthcare professionals and then automatically uploaded to the database.

By continuously monitoring a patient, wearable health devices can provide a more comprehensive view of a patient’s health status. However, it is important to ensure that the system is secure, respects the patient’s privacy, and complies with relevant regulations and standards [64,65]. However, it has been observed that wearable gadgets are frequently taken off and thrown away for either purposeful or unintentional reasons, so a balance needs to be struck between functionality, dependability, and cost. This is a common issue with wearable health monitoring devices, particularly among patients with dementia, who may forget where they have placed their device or may not understand the importance of wearing it consistently.

Non-contact monitoring of vital signs using cameras and image recognition techniques is a promising area of development in healthcare technology and has the potential to improve the accessibility, efficiency, and cost-effectiveness of vital sign monitoring. The use of AI-based image recognition algorithms, mainly deep learning architectures, allows images to be automatically analysed to assess vital signs.

YOLOv3 (You Only Look Once, Version 3) [66] is a real-time object detection algorithm that allows specific objects to be identified in videos. YOLOv3 uses a variant of the Darknet neural network architecture, specifically Darknet-53 as its backbone network. The architecture consists of 53 convolutional layers, which was trained on the ImageNet dataset, which was designed for computer vision research [67]. YOLOv3 also contains several key features that help to improve the detection accuracy and performance, including residual skip connections, upsampling, and multiscale detection. The most important feature of the algorithm is that it performs detection at three different scales by downsampling the dimensions of the input image by factors of 32, 16, and 8, respectively (see Figure 2).

The AlphaPose algorithm allows us to detect keypoints in the bodies of several people with high accuracy in real-time video or images. The 17 keypoints detected by AlphaPose include the nose, eyes, ears, shoulders, elbows, wrists, hips, knees, and ankles (see Figure 3). As the Figure 3 shows, the algorithm can successfully detect the following keypoints in video footage of a patient in a movement position. All of these keypoints are used to construct a human body skeleton representation, which can be used for various applications such as activity estimation [68], process recognition [69], and human fall detection in different environments [54,55,70,71].

In particular, a decision support system relies primarily on the expert knowledge of geriatric staff nurses who are experienced in developing nursing plans for patients with different health problems. Their expertise has been used to create the rules that guide the decisions made by nursing professionals which, in this case, are mapped into the output of how to proceed with the nursing plan. Individual experts suggest different decisions based on critical factors in certain cases, so it would seem reasonable to use Fuzzy logic or Neuro-fuzzy models, which are more similar to human thinking. However, given that most of the input variables are of the verbal and integer type, the use of such models will not be efficient. In addition, we do not have enough statistical data to create mappings between numerical values and verbal estimates (e.g., Breathing: Increased → *X* breaths per minute) and to create fuzzy sets based on this. Therefore, we decided to rely on the Decision Tree supervised learning approach which can handle both numeric and non-numeric values, has fast decision times, enables parameter optimisation, and has the possibility of refinement if the accuracy of the result is not satisfactory (e.g., Random Forest). In the decision support module, a Decision Tree with a Gini impurity value was used, and a prepruning process was applied to prevent overfitting. The Gini impurity value is given by
$$Gini=1-\sum_{j=1}^{c}p_{j}^{2},$$
where $p_{j}$ is a proportion of observations that belong to class *c* for a particular node.

The fine-tuning of Decision Tree hyperparameters involves a depth limited to a maximum of 3 and a minimum number of samples equal to 6 in a finite node. An average classification error of 92% was achieved.

For patient reidentification, the study made use of the Bag of Visual Words (BOVW), since it has been proven to be successful in a number of computer vision tasks, including human reidentification and human action classification [72,73,74]. With the BOVW approach, local features (such as SIFT descriptors) from images are first extracted and then grouped into a visual vocabulary. Each image is then represented as a histogram of visual words, which may be used for classification or retrieval tasks using machine learning algorithms. More specifically, the K-means algorithm was trained using the final list of features that were retrieved from patient images. As a result, the features were grouped into visual words. Finally, a ML-based classifier was used to generate a categorisation of images based on a newly created vocabulary.

### Performance Metrics

The *F*1 score is a metric that is widely used to evaluate the performance of a classification model. For a multiclass classification, the *F*1 score for each class is calculated using the one-against-rest (OvR) method. In this approach, the metric for each class is determined separately. However, rather than assigning multiple *F*1 scores to each class, it is more common to take an average and obtain a single value to measure the overall performance. Three types of averaging methods are commonly used to calculate *F*1 scores in a multiclass classification, but only two of them are recommended for unbalanced data, as in our case. More specifically, macroaveraging calculates the *F*1 score for each class separately and derives an unweighted average of these scores. This means that each class is treated equally, regardless of the number of samples it contains. The macroaveraging *F*1 score is given by
$$Macro_{avg}\;F1=\frac{\sum_{i=1}^{n}\;F1_{i}}{n},\;$$
where *n* is the number of classes. In contrast, a weighted averaging calculates the *F*1 score for each class separately and then takes the weighted average of these scores, where the weight for each class is proportional to the number of samples in that class. In this case, the *F*1 result is biased towards the larger classes, i.e.,
$$Weighted_{avg}\;F1=\frac{\sum_{i=1}^{n}\;w_{i}\times F1_{i}}{n},\;$$
where $w_{i}=\frac{k_{i}}{N}$ is the weight of the class *i*, *N* is the total number of samples, and $k_{i}$ is the number of samples in the class *i*.

## 4. Results

### 4.1. Implementation of the Geriatric Care System

The geriatric care plan system for end-users, i.e., nursing home staff, was created using C# programming language and the ASP.NET Core 6.0 framework for the back-end. The front-end was built using Node.js version 19 and the Angular framework, while testing was carried out using Karma. PostgreSQL was used as an open-source relational database management system. The use of these technologies allowed developers to create a robust and scalable system that was able to handle the large amounts of data generated by IoT devices. In addition, Docker was used to containerise the software for deployment by combining the system and all of its dependencies into a single container that could be quickly deployed on any platform that is compatible with Docker. The architecture of the system is demonstrated in Figure 4.

Wearable gadgets synchronise the data with cloud servers, since the data they generate needs to be processed and analysed. Once the data have been received by the cloud servers, the company’s server pulls the data from the Google cloud servers using API and then parses the files and saves information in the Postgre database. In contrast, the data captured by the cameras are sent directly to the server. This dataset is then processed in the back-end and analysed alongside the wearable data in order to provide a more comprehensive view of a patient’s health status. The main purpose of .NET backend is to act as a bridge, passing data between the Angular front-end and the Postgre database. The back-end is written using REST API methodology to provide a standardised way for different applications or devices to communicate.

### 4.2. AI-Based Data Analytics and Decision Making

To prepare a nursing care plan, a rich set of data is collected about the patient, as summarised in Table 2. Then, the recommendations for the actions to be taken in a nursing plan are generated from the geriatric care management system.

For demonstration purposes, the collected data were analysed to detect possible dependencies. The radar graph below (see Figure 5) is a single patient’s chart of selected vital signs over a 50-day observation period, displaying SBP (systolic blood pressure), DBP (diastolic blood pressure), HR (heart rate), SPO2 (oxygen saturation), sleeping hours, and weight measurements. The data analysis was carried out on three patients on the ward, but no significant dependencies between variables were identified. It may be assumed that that some trends could be determined if the data were gathered over a longer period of time and additional variables, such as pain level, temperature, and even verbal type indicators, were included.

The geriatric care personnel was responsible for writing the rules for the care plans. These rules were based on best practise and experience in the field and were designed to ensure that patients receive the most appropriate and effective care. More specifically, care plans were tailored to the specific needs of current and future patients, taking into account their medical history, current condition, and other relevant factors. The variables listed in Table 2 were included in the care plan, as they indicate the patient’s medical history, current health status, and other relevant factors that can influence treatment. On the basis of this information, the initial set of rules covered a wide range of scenarios and options, but after optimisation, the patient care plan eventually consisted of 61 rules with the four possible outputs of the care plan: “Continue current treatment”, “Monitor”, “Adjust”, or “Extra situation” (see Figure 6). All remaining cases that were not included in the rules were assigned to the care plan “Continue current treatment” by default.

In particular, the nursing care plan was designed to be flexible and adaptable to allow healthcare professionals to adjust the patient’s geriatric care according to his or her health status and changing needs. Those rules and the output generated by the geriatric care management system help healthcare personnel to respond more quickly to changes in a patient’s health, shape the patient-personalised geriatric care, reduce the risk of human error, and make better use of staff time by concentrating more on essential social support.

Figure 7 shows a schema for an AI-based decision support system. Four of the 21 variables (see Table 2) are automatically registered; that is, three of them were retrieved from IoT devices and one (change in movement) was obtained from the camera. The value of the latter variable was generated from the AI-based image recognition module. The remaining variables were taken from the MS Excel spreadsheet file, where all data were entered manually.

#### Image Recognition Solution

An AI-based image recognition module is a block consisting of several sequential algorithms that detects changes in the movement of a patient. In this project, we used the camera to film nursing home patients, that is, one room with three patients. The video was recorded at 1920 × 1080 pixel resolution with a frequency of 10 FPS, therefore storing 10 unique images per second to obtain 10FPS × 60 = 600 images per minute. An image was analysed every five seconds with the assumption that no significant changes in motion would be detected in that time period.

The image processing included

Brightening: to increase the overall luminosity of the image, improve visibility, and increase the clarity of the image during low light conditions;Cropping: to keep only regions of interest in the image;Denoising: to remove noise from the image, typically by applying a low-pass filter. It also improved the quality and clarity of the image by removing noise, which could be especially useful if the image was taken under poor conditions or with a low-quality camera.Edge detection: to identify edges in the image by finding points of a rapid intensity change. It can also be used to identify and extract features or objects in the image, such as lines, shapes, or boundaries.

After image processing, the algorithm integrating YOLOv3 and AlphaPose [75] was used to detect human poses. The algorithm includes the three main components [76]. First, the Symmetric Spatial Transformer Network (SSTN) takes the detected bounding boxes to generate pose proposals. The SSTN allows the spatial context and correlations between the keypoints to be captured, leading to more accurate pose estimates. Second, the Parametric Pose Non-Maximum-Suppression (NMS) is a component that is used to remove redundant pose detections and improve the overall accuracy of the pose estimation. Finally, the Pose-guided Proposals Generator is used to create a large sample of training proposals with the same distribution as the output of the human detector.

The next step is the problem of identifying and classifying patient postures, which in this case, included the following six postures: “walking”, “standing”, “sitting”, “fallen on the ground”, “lying in bed”, and “sleeping”. For the verification of all poses, a sequence of three images was taken for a period of 15 s, except for the last two poses. The poses of “sleeping“ and “laying in bed“ correlate with the parameters of the smart bracelet (sleep time and heart rate), so these parameters were also assessed. If the patient was found to be lying in bed, the assessment time was extended by up to one minute to identify whether the patient was “lying in bed” or “sleeping”. In particular, the pose was assessed every minute until a new pose was captured.

A pose change algorithm was developed to detect differences between adjacent images, that is, to identify that a person was walking rather than standing or that a person was just lying on a bed rather than sleeping. Figure 8 illustrates the example of three iterations of assessment frames of the “walking” pose taken every five seconds. Comparing the images taken every five seconds, we can see that the pose remained the same, although the frames were not identical and the patient’s coordinates varied.

In order to define changes in movement habits, an additional algorithm (see Algorithm 1) was created to evaluate movement changes over a longer period of time $t_{m-n}$, where *m* is a current time moment, *n* is a number of days before $t_{m}$, 1 ≤ n ≤ 3. This algorithm calculates the duration (hours) in each pose per day. The percentage change is then evaluated, compared with threshold value $k_{th}$ and a response is generated that includes three possible values: “Unchanged”, “Slowed down”, or “Increased”. The pseudocode of the algorithm is provided below.
**Algorithm 1** Evaluation of changes in movement habits$Act_{H}=Walking\left(hrs\right)+Sitting\left(hrs\right)+Standing\left(hrs\right)$$k_{th}=12.5\%$Diff(a,b)=((a−b)/b)*100**if** $Diff(Act_{H}\left(t_{m-1}\right),Act_{H}\left(t_{m-2}\right)\leq -k_{th}\wedge Diff(Act_{H}\left(t_{m-2}\right),Act_{H}\left(t_{m-3}\right)\leq -k_{th}$ **then**    MoveH is slowed down**else if** $Diff(Act_{H}\left(t_{m-1}\right),Act_{H}\left(t_{m-2}\right)\geq k_{th}\wedge Diff(Act_{H}\left(t_{m-2}\right),Act_{H}\left(t_{m-3}\right)\geq k_{th}$ **then**    MoveH is increased**else** MoveH is unchanged**end if**

For demonstration purposes, the identification of tough poses observed in the real-world environment is shown below. For instance, Figure 9 shows a skeleton-based posture recognition in various lighting environments. In well-illuminated areas, patients can be detected by identifying all skeleton keypoints (Figure 9c). It has been observed that at night or at twilight/night, walking patients can be identified quite accurately with all keypoints (Figure 9a,b), but when patients are sleeping with their blankets, few keypoints were successfully detected (Figure 9d) or keypoints were not detected at all (Figure 9a).

Another example demonstrates a skeleton-based posture recognition for two different scenarios. In Figure 10a, the keypoints in the patient’s body were detected when the patient was lying on the ground, which refers to the status “falling on the ground”. To correctly recognise this pose, a training dataset with artificially simulated falling poses was created. Comparatively, Figure 10b shows that the keypoints in the body were identified for all persons located in the ward, but the nursing personnel needed to be the exception. Therefore, additional data were collected to train a deep learning algorithm to distinguish staff from patients. Consequently, the nursing personnel was identified by their clothes, more specifically, white trousers and a blue top, which they had to always wear.

### 4.3. Experimental Results

A posture detection algorithm of captured video material was tested to identify six different poses. The results are summarised in a confusion matrix to evaluate the performance of the algorithm. More specifically, the confusion matrix provides a visual representation of the number of correct and incorrect predictions made by the classifier: the rows represent the actual class labels, while the columns represent the predicted class labels. The diagonal elements show the number of correct predictions (see Figure 11).

The posture recognition algorithm was trained using 9300 labelled images and tested using 3792 images. An average posture recognition accuracy of 91.63% was achieved for the testing data set (Figure 11). Posture labelling was performed manually on the images obtained from the video stream for training and testing purposes. The Receiver Operating Characteristic (ROC) curve of the stratified testing dataset is provided in Figure 12.

The AUC values for each posture class ranged from 0.8790 to 0.9427, with the highest value obtained for the sitting posture class. The sleeping and lying in bed posture classes resulted in the lowest AUC values, with values of 0.9047 and 0.8790, respectively. These lower values suggest that it might be more difficult for the classifier to distinguish between these postures and others. Comparatively, the AUC value for the fallen on the ground posture class was 0.9177, which is slightly lower than those of the other more successfully recognised posture classes. This could be due to the lack of training data for this posture, which might have led to lower accuracy. Next, Table 3 summarises the estimated values of precision, recall, and *F*1 score for each class of interest, together with macro and weighted *F*1 scores for the evaluation of the overall performance of the posture recognition algorithm.

The patient re-identification testing results are summarised in Figure 13. The support vector machine (SVM) method was used to generate categories of images, providing labels for the patient classes. In our case, the maximum number of classes was set to four: three classes represented the maximum number of patients the ward can accommodate, while the separate class “None” referred to unauthorised individuals such as nursing staff, family members, doctors, or others. The class names for patients were labelled “First”, “Second”, and “Third” (see Figure 13).

The confusion matrix in Figure 13 summarises how successfully the algorithm identifies three ward patients in common areas. One can observe that an accuracy level of 90% was obtained for the “First” class, a value of 88% was obtained for the “Second” class, and a value of 91% was obtained for the “Third” class. Although the lowest accuracy level of 87% was achieved in the "None” class, considering that there can be around 6–13 people in a single tray, this is a pretty good accuracy level. It was observed that female patients and nursing home staff were more easily recognised, but other patients, nonmedical nursing home staff, and visiting relatives were the most confused with these patients.

Finally, to conduct a real-time experiment, patient positioning verification was carried out. This included 16 scenarios with diverse positions. The results are summarised in Table 4. Two prediction errors were determined. More specifically, the patient was “lying in the bed”, but he was detected as “sleeping”, as he was covered up, his heart rate was reduced, he did not move for more than one minute, and it was night time. Another prediction error also related to the sleeping pose. The patient was lying in the bed without covering up; however, it was determined that he was not sleeping based on readings from the smart wristband. It should be noted that the prediction may also be impacted by ambient light conditions. From a technical perspective, the proposed system performed pose estimation with an average output time of 182 ms, including the algorithm used to predict the pose from the possible outcomes.

To test the correctness of the output of the geriatric care management system, different scenarios involving nursing home staff were developed. The results revealed that the system provided the correct output in all cases. The system was designed to generate changes to the treatment plan immediately after any changes are made. When a healthcare professional makes a change to the care plan, the system analyses the data from the patient’s IoT devices and determines the appropriate course of action. The system then automatically updates the results of the action to be taken for the individual patient and alerts the healthcare professional. This allows healthcare professionals to stay up-to-date with the patient’s condition and make any necessary adjustments to their treatment in a timely manner.

## 5. Discussion

There are a few areas for improvement, as the proposed geriatric management care system is still in its initial stage of functioning. Personality identification, which relates to the continuous contactless assessment of the patient, is the most challenging concern. Comparatively, wearable devices do not raise any questions at the moment; their purpose is clear, but elderly people have a problem with wearing them because they find them annoying. The creation of the nursing care plan itself could be fully automated later on, with a follow-up on what action should be taken when the situation changes. However, to fully automate it, a lot of statistical data are needed, including actions taken by nurses, from which the system could be learnt, that is, from the actions taken by the care worker on each individual situation. Taking into account the current data (Table 2), there are at least 20,155,392,000 possible combinations of parameters that define the health condition, which are likely to increase in the future due to the inclusion of additional parameters. For this purpose, a list of actions is provided in the geriatric care management system, from which the care worker must indicate (select from the list) what they intend to do. In this way, a dataset of situations and decisions with all the actions taken accordingly is continuously accumulated. Once a representative sample of data has been accumulated (say after at least one year), the correctness of the automated action is improved.

The challenge with consistent and accurate patient identification makes it reasonable to consider other methods of individual identification than BOVW. As patients usually stay in their own wards, the accuracy of identification is high when the patients are present and nursing staff visit them a certain number of times per day. However, the accuracy drops in common areas (e.g., resting, eating) because there are more patients and personnel present. Mainly because of their distinguishing clothes, nursing workers are simpler to identify (see column “None” in Figure 14). However, the elderly patients themselves are more likely to be confused with each other in common areas, with a best individual identification result of 0.914 achieved (see Figure 14).

As an alternative method, gait recognition (GR) technology can be used for patient identification. This method examines the uniqueness of an individual’s walking or running pattern using machine learning (ML) techniques [77]. More specifically, ML algorithms are trained to recognize subtle differences in a person’s gait and thus can use this information to identify individuals even if their face is obscured in the image [78,79]. An additional benefit of GR technology is that gait information can be used not only for personal identification, but also for medical purposes, such as monitoring and for the diagnosis and treatment of various movement disorders [80,81]. For example, gait recognition technology can be used to identify and diagnose various types of neurodegenerative diseases (such as Parkinson’s and Alzheimer’s disease) or assess the course of disease [82,83,84,85]. This can help doctors and healthcare professionals to develop more effective nursing care plans and interventions as well as monitor the progression of these conditions over time. However, GR technology usually requires a variety of sources or capture devices to gather data about an individual’s gait, including multiple video cameras, motion sensors, radars, and other specialized equipment [79,86]. In addition, the accuracy of gait recognition technology can be affected by a range of factors, including the angle at which the gait is captured.

Finally, it should be noted that elderly people are choosing to live independently at home for as long as possible. In such cases, intelligent geriatric care management system monitoring adapted to the individual home and operating remotely can be very helpful for ensuring that the elderly person is safe and providing faster reactions to emergencies (i.e., fall detection) and appropriate care. In the near future, we plan to develop the necessary software and hardware package (e.g., for the proper functioning of the system such as a stable internet connection) for the home care services and to test it in real-world environment with the possibility of transmitting the data to the responsible physician for monitoring.

## 6. Conclusions

In this study, a geriatric care management system based on IoT and AI algorithms was proposed to monitor some of the most important vital signs in a noncontact manner and to facilitate the adjustment of the care plan. The system provides an intelligent assistance function, which suggests how to proceed with the patient’s care plan based on the available data and the decision support module.

A built-in posture recognition algorithm allows staff to react quickly to extreme situations, which are highly expected at night or during peak working hours. Another algorithm was developed to monitor changes in a patient’s movement habits over a longer period of time, which can be important for detecting health problems more quickly and taking appropriate action. This is a value-added functionality of the system, as it is very difficult for nursing staff to do this in a natural way, as it is not possible to monitor every patient 24 h a day without smart technology. During this study, it was observed that the most confusing poses are “lying in bed” and “sleeping”. Detecting the individual or pose when the patient is fully or partially occluded is also quite challenging. However, capturing the pulse and sleep mode and combining these indicators with the outputs of the image recognition algorithms resulted in better detection of the “sleeping” and “lying in bed” poses, i.e., the accuracy was improved by around 15.48% and 22.06%, respectively. Additionally, the system is resistant to data deficiencies; if certain data are not received at the current time, the value is taken from the last time of recording. In any case, the final decision is made by the human, and in case of error or incorrect output, one has the opportunity to correct it.

Other concerns are ensuring that smart health monitoring devices are worn and maintained at all times, as patients often want to remove devices (particularly patients with a dementia), and nursing staff do not always notice quickly when the devices need to be loaded. Therefore, the involvement of care specialists is crucial to ensure the system operates effectively and efficiently. In addition, it is equally important to make sure that patients feel comfortable and moreover that their privacy and trust in smart technologies are maintained at the appropriate level. By involving nursing staff in the implementation process, they can provide valuable feedback, suggestions, and ideas, leading to a better overall outcome.

## Figures and Tables

**Figure 1 healthcare-11-01152-f001:**
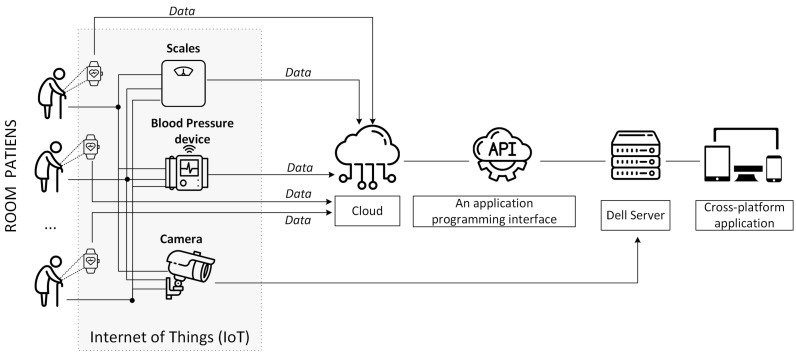
Data collection pipeline of the GCM system.

**Figure 2 healthcare-11-01152-f002:**
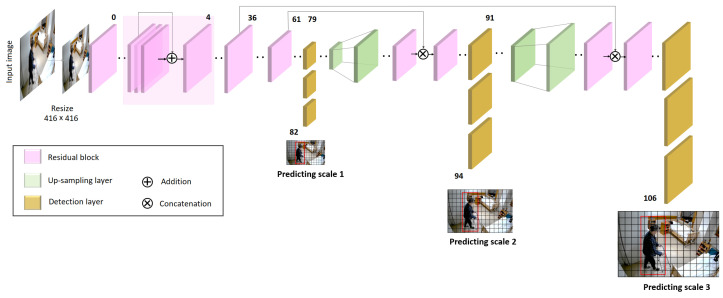
The architecture of YOLOv3 algorithm.

**Figure 3 healthcare-11-01152-f003:**
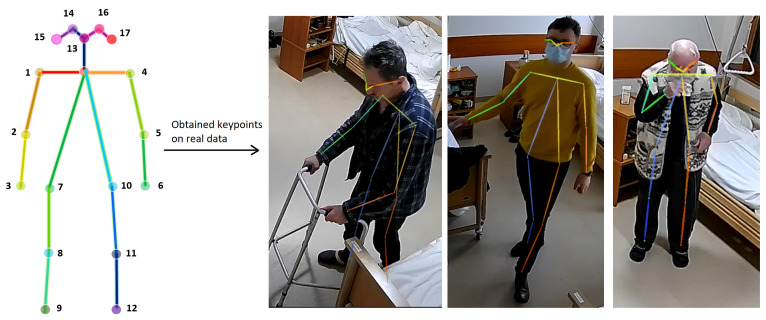
AlphaPose algorithm illustration: keypoints on patients’ bodies in video footage.

**Figure 4 healthcare-11-01152-f004:**
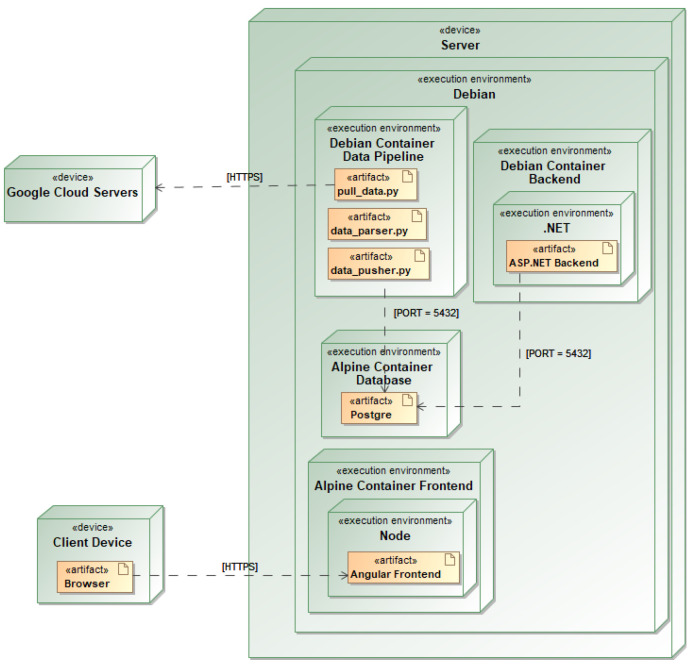
UML deployment diagram of the geriatric care system architecture.

**Figure 5 healthcare-11-01152-f005:**
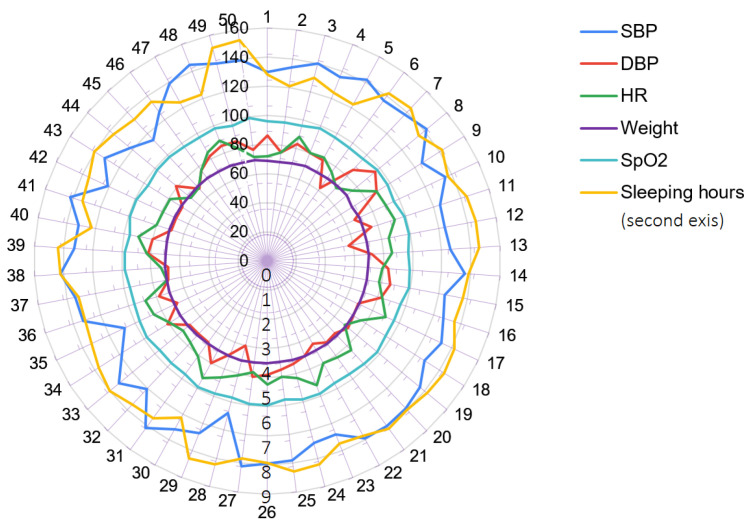
Six different health parameters collected for a single patient.

**Figure 6 healthcare-11-01152-f006:**
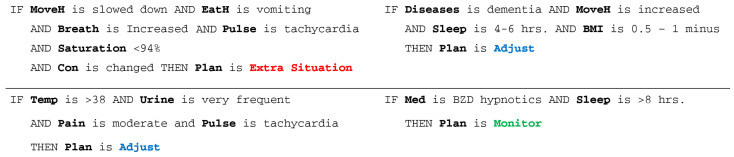
Examples of different care plan with IF-THEN rules defined by the staff.

**Figure 7 healthcare-11-01152-f007:**
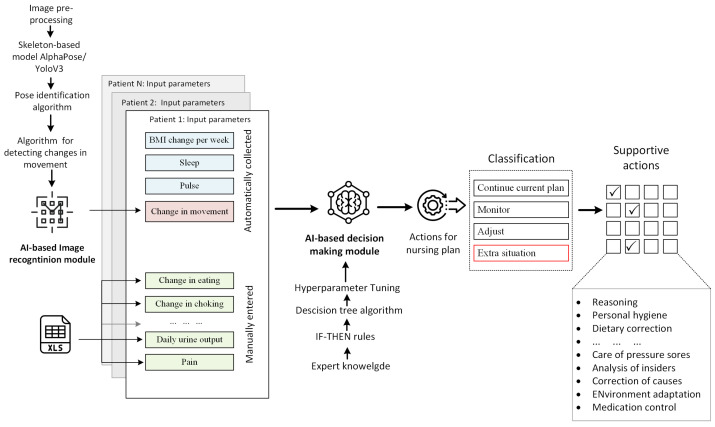
Schematic diagram of proposed geriatric care management systems.

**Figure 8 healthcare-11-01152-f008:**
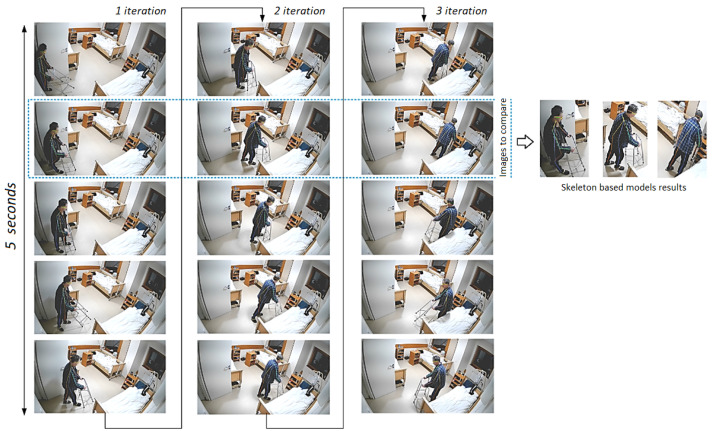
Three iterations of the assessment frames of the patient in the “walking” pose taken every five seconds.

**Figure 9 healthcare-11-01152-f009:**
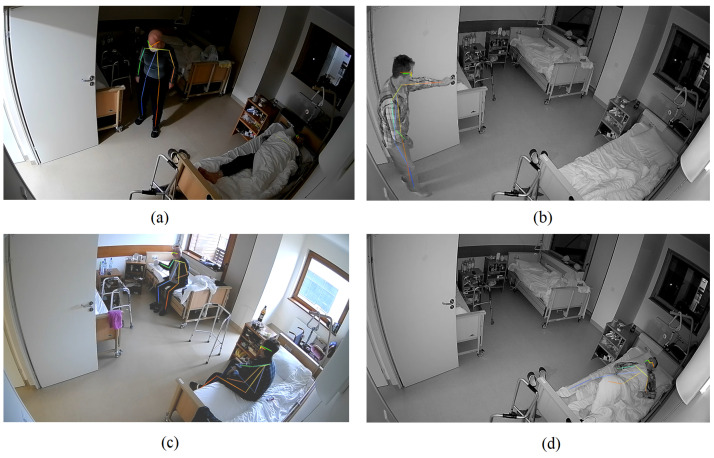
Examples of skeleton-based posture recognition in various ambient light conditions: (**a**) patient walks in a semi-lit environment; (**b**) patient walks during the night; (**c**) two patients sit in a fully-lit environment; (**d**) patient is lying down at night.

**Figure 10 healthcare-11-01152-f010:**
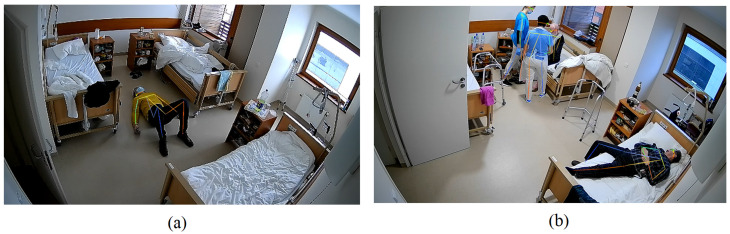
Examples of skeleton-based posture recognition in different scenarios: (**a**) the patient is lying on the ground; (**b**) patients are visited by nursing staff.

**Figure 11 healthcare-11-01152-f011:**
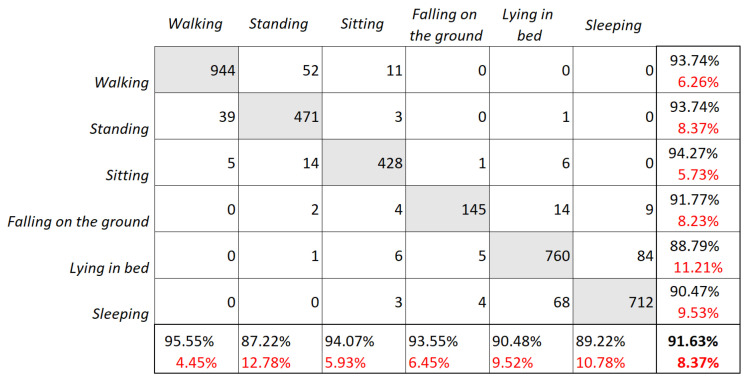
Testing results: confusion matrix of posture classification.

**Figure 12 healthcare-11-01152-f012:**
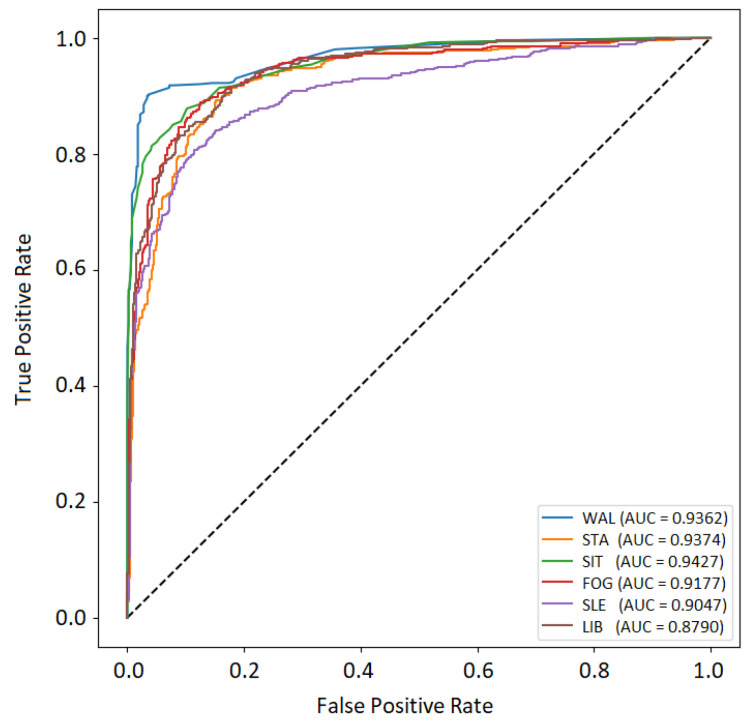
Testing results: ROC curve of the posture recognition algorithm.

**Figure 13 healthcare-11-01152-f013:**
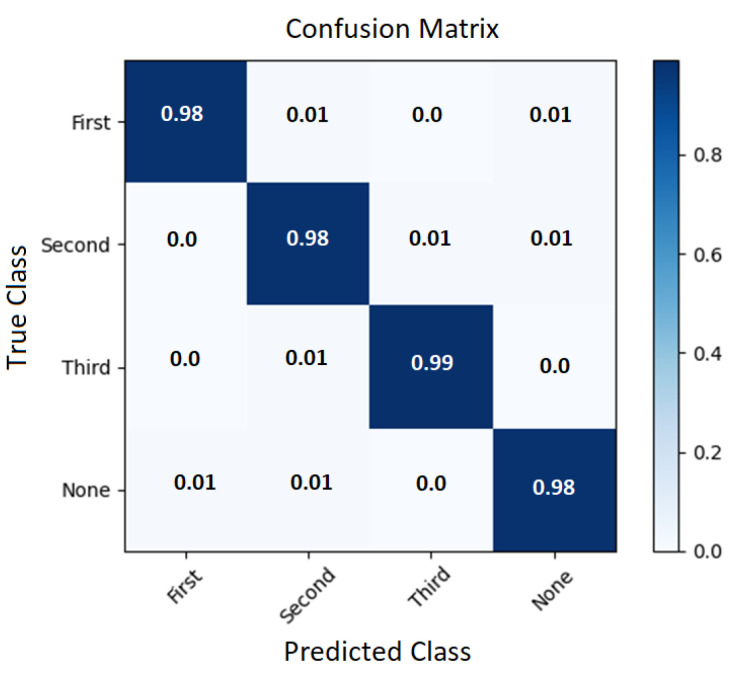
Confusion matrix showing the re-identification of three patients (referred to by the class labels “First", “Second", and “Third".)

**Figure 14 healthcare-11-01152-f014:**
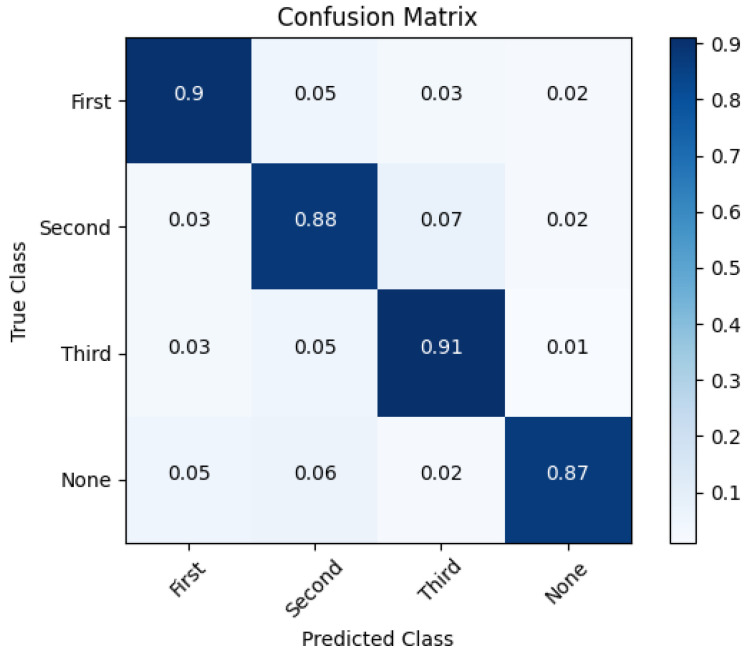
Confusion matrix of the re-identification of three patients (referred to class labels “First”, “Second”, and “Third”) in the two common areas of nursing homes.

**Table 1 healthcare-11-01152-t001:** Types of IoT devices used in the research.

IoT Device Type	Device Name
Camera	EZVIZ CS-C3TN 1920 × 1080
Wrist band	Fitbit Charge 5
Blood Pressure	Withings BPM Connect
Scales	Withings Body+

**Table 2 healthcare-11-01152-t002:** Patient information.

No	Variable	Definition	Instances of Possible Values/Range
1	FN	First name	-
2	LN	Last name	-
3	BD	Birth date	yyyy/mm/dd
4	HE	Height	1.20 m–2.20 m
Input data			
1	MoveC	Movement capabilities	Lying; sitting in a wheelchair; with assistive devices; etc.
2	RiskC	Risk of collapse	None; low; medium; high
3	Bedsores	Bedsores	Yes; no
4	Diseases	All patient’s diseases	Heart failure; Alzheimer; dementia; Cancer; etc.
5	Med	Taken medications	Antibiotics; antihypertensives; antidepressants; etc.
6	BMI	BMI unit change per week	<0.5 plus; <0.5 minus; 0.5–1 plus; etc.
7	MoveH	Movement habits	Unchanged; slowed down; increased; falling on the ground
8	EatH	Eating habits	Parenteral nutrition; fed by another person; independent eating; etc.
9	EatC	Eating capabilities	Swallows solid food; swallows only mashed food; swallows only liquids; etc.
10	Bowel	Bowel habits	Regular bowel movements; diarrhoea; constipation; faecal incontinence
11	Sleep	Sleeping	<4 h; 4–6 h; 6–8 h; >8 h; apnoea
12	Breath	Breathing	Increased; slowing down; with apnoeas
13	PL	Pulse	Normal; bradycardia; tachycardia
14	BP	Blood Pressure	Normotension; hypotension; hypertension mild; hypertension moderate; hypertension severe; etc.
15	Temp	Temperature	<36.0 °C; 36.0–37.4 °C; etc.
16	Sat	Saturation	≥94%; <94%
17	Urine	Daily urine output	Concentrated urine; very frequent; etc.
18	Fluid	Fluid tracking	<500 mL; ≥500 mL
19	Gly	Glycaemia	<2.5 mmol/l; ≥2.5 mmol/l
20	Con	Consciousness	Unchanged; changed; unconscious
21	Pain	Perceived level of pain	None; mild; moderate; severe; unbearable
Output data			
1	Plan	Nursing plan	Continue current plan; monitor; adjust; extra situation

**Table 3 healthcare-11-01152-t003:** Testing results: performance metrics of the posture recognition algorithm.

Class	Precision	Recall	*F*1 Score
Walking (WAL)	0.9554	0.9374	0.9463
Standing (STA)	0.8722	0.9163	0.8937
Sitting (SIT)	0.9406	0.9427	0.9416
Fallen on the ground (FOG)	0.9354	0.8333	0.8814
Lying in bed (LIB)	0.8951	0.8878	0.8914
Sleeping (SLE)	0.8844	0.9047	0.8944
	Macro *F*1 score		0.9082
	Weighted *F*1 score		0.9125

**Table 4 healthcare-11-01152-t004:** Real-time scenario testing results of posture recognition.

No.	Actual Pose	Predicted Pose	Ambient Lighting	Confidence
1	Walking	Walking	Day time (well-lit)	98.0%
2	Sitting	Sitting	Day time (well-lit)	97.5%
3	Sitting	Sitting	Day time (well-lit)	98.2%
4	Lying in bed	Sleeping	Night time (poorly lit)	89.3%
5	Standing	Standing	Day time (perfect)	99.7%
6	Lying in bed	Lying in bed	Evening time (semi-lit)	87.9%
7	Sleeping	Lying in bed	Evening time (semi-lit)	88.6%
8	Standing	Standing	Day time (perfect)	99.1%
9	Sleeping	Sleeping	Night time (poorly lit)	85.4%
10	Walking	Walking	Day time (perfect)	93.6%
11	Lying in bed	Lying in bed	Day time (perfect)	94.2%
12	Standing	Standing	Day time (perfect)	99.3%
13	Walking	Walking	Night time (poorly lit)	96.0%
14	Sitting	Sitting	Day time (perfect)	98.5%
15	Sleeping	Sleeping	Day time (perfect)	91.0%
16	Fallen on the ground	Fallen on the ground	Day time (perfect)	99.8%

## Data Availability

The data are not publicly available due to privacy restrictions.

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
