# Peer review of "Geriatric Care Management System Powered by the IoT and Computer Vision Techniques"

_healthcare, 2023, doi:10.3390/healthcare11081152_

Round 1

Reviewer 1 Report

In this paper, the author addresses the shortcomings of the existing geriatric care management system and proposes an innovative design that utilizes wearable sensors and non-contact measurement technology. By using artificial intelligence algorithms, the proposed system is able to detect abnormal health status in patients and generate and update care plans accordingly. However, there are still some problems. Please refer to comments.

  1. Is it necessary to include evaluation indicators or conduct comparative analysis for the action part of the nursing plan generated by the proposed digital elderly care plan system?

  2. Can the proposed system adjust the monitoring function and nursing plan for special patients or the elderly?

  3. Some of the pictures in the article are unclear, and the text in the pictures is too small.

  4. Some of the sentences in the article are not smooth. For example, the sentence "First, elders' capacity to acknowledge the need of and properly use weapons positions new challenges" in line 178 could be rephrased to improve clarity.

Author Response

Please see the attachment, where the response letter and the revised manuscript combined to a single file can be found.

Reviewer 2 Report

The topic of the work is relevant and the need for these approaches is well known. The real context being addressed by the authors is also valuable and the overall idea presented in the abstract and introduction makes sense and has merits. There are, however, several aspects in how the work is presented that do not allow understanding the actual contributions the authors are aiming to disclose. Please consider, without the pretense of thoroughness, some comments and suggestions that inform my perspective:

* One aspect that needs to be clarified concerns the goal for the work and what is presented. In line 133 the authors write: "...we propose an intelligent system for digitizing the care plan...". I assume that the care plan refers to the nursing care plan. However, to the best of what I could understand no contribution is made to that particular aspect, since the overall contents of the article concern monitoring. I would suggest that the authors establish their focus more clearly. The Related Work, for instance, addresses monitoring, with a strong emphasis on fall detection, and no advanced perspective on the nursing plan and related goals and tasks is explored. Then, around pg 10, the nursing staff gains prominence, again, with participation in rule definition, etc. Nevertheless, the only evidence provided is the pose detection method being presented, evaluated, and discussed. The authors probably need to more clearly define what is actually their contribution and whether they revise the abstract and introduction or provide stronger evidence for the overall claim.

* From my perspective, the introduction is probably longer than needed, mixing several aspects or providing additional detail that could go in the Related Work

* pg 2 ln 37, the times for the different aspects are given without any kind of support

* details on the study mentioned in lines 40 and beyond is important as this is given as justification for the requirements

* pg 3, ln 129, the nature of medical devices is briefly discussed, but the authors might, in a clear way, enunciate what thei are aiming for; while the digitized care plan is assumed as something not having a medical device nature, the following paragraph talks about detecting abnormal health status.

*ln 251, "a common solution is" -> what are the grounds for this statement?

* It seems that section 3 is not something concrete regarding the authors' work and reads more like additional background and related work

* ln 325 "this allows.. treatment plans to be adjusted in real time" -> where is the evidence for this?

* ln 365, figure 2 is described as presenting data for a single patient; the caption, how it is phrased my induce in error. Maybe "Six different health parameters for one of the monitored patients"?

* ln 380 - 390, same text is repeated

* sec 4.2.1, if the authors are using the off-the-shelf version of YOLO, it might make no sense to be explaining its inner pieces and presenting its architecture.

* As far as I could understand, the authors do not discuss how they deal with the fact that three patients are present in the room and how they deal with data separation and patient identification to attribute data;

* ln 476, falling is briefly alluded and, then, reapears in line 489. Maybe unify content to improve clarity? And why, if the person sits, before a period of 20 seconds after falling the situation is no longer "extra situation"? Isn't falling a stong indication of a potentially critical situation involving aspects such as dizziness, etc?

* And "falling on the ground" is described as one of the poses detected in line 489; however, it is not part of Table 4; any reason for that, since it was tested for the results presented in table 3?

* to the best of what I could grasp, nothing is evidenced about any kind of data being provided to the nursing staff, e.g., in line with a monitoring dashboard?; the clains of "a system designed to be intuitive and user-friendly" present in the abstract are not supported

* the architecture presented in  figure 12 seems to relate to the data collection and processing, but nothing is provided for a more global architecture of the system as supporting the nursing plan

* Even though much data seems to be collected and input by the nursing staff, nothing is evidenced about how the system fulfills the purpose of signaling critical changes or situations; the rules mentioned in figure 3, how are they input into the system? how does the nursing staff control these aspects? What is the user-interface made available to the staff? The text alludes to the nursing staff being involved in a "care flow", e.g., lines 580-588, but not a single evidence of these is presented and discussed.

* The pose estimation method is tested for data collected from three patients, if I am not mistaken; such low number of patients probably requires a discussion of how this might impact the interpretation of results? How much data is used to obtain the results presented in Table 4? Where does the actual pose labels come from?

* The discussion points to the challenge being addressed as pose detection, which describes the work as just a small piece of the initial "promise" of the abstract and introduction.

* As a general comment, I would suggest that figure captions are revised; for some of them, the caption does not provide any context for what is being presented. I would suggest, as a good practice, to not make figure interpretation completely dependent on the text.

Author Response

(The authors gave the same response as above.)

Reviewer 3 Report

The topic is interesting, timely and potentially clinically important. The reporting and writing is of high quality. The only comment I have is in regard to line 91 where references are missing. Also, given wearable sensors are core to the paper, I would advise emphasizing some of their additional utilities for example joint angle estimations https://www.ncbi.nlm.nih.gov/pmc/articles/PMC3175383/, activity classification https://www.nature.com/articles/s41598-022-18845-x, fall prediction https://www.sciencedirect.com/science/article/pii/S2665917422002483

Author Response

(The authors gave the same response as above.)

Reviewer 4 Report

Digitisation of geriatric care plans refers to the use of technology to manage and deliver care to elderly individuals involving electronic health records and mobile apps to streamline the care process and improve the quality of care for seniors.

The aim of the authors is to develop an intelligent system for digitising a care plan, combining signals from various wearable sensors and non-contact measurement techniques, and employing AI algorithms to detect abnormal health status of a person.

Their  system is based on the Internet of Things (IoT) and computer vision algorithms for the automated collection of patient health data. In addition, the proposed algorithm allows monitoring changes in the patient's position over a longer period of time, which can be important for detecting health problems more quickly and taking appropriate measures. The proposed system is designed to be intuitive and user-friendly, making it easy for caregivers and healthcare professionals to use and monitor the health and well-being of patients.

The study needs revisions before being considered for acceptance.

Strengths:

The work  is attractive and very interesting

It is developed with great enthusiasm

Points of weakness

Little attention to the editorial structure

Little attention to small details, such as, for example, the description of the figures, the images, etc.

Further comments:

1.       The abstract must be rewritten to better summarize the sections

2.       Please better explain the purpose “In this study, we propose an intelligent system for digitising a care plan, combining signals from various wearable sensors and non-contact measurement techniques, and employing AI algorithms to detect abnormal health status of a person. The..” If needed, you can expand with bullet points.

3.       Methods: insert a scheme or sketch of flow chart describing the design.

4.       Sections 4 and 5 are results. I suggest to rearrange them into a section with two sub sections. The first is dedicated to the system, while the second is dedicated to testing/validation of the system

5.       Discussion is not a discussion. You must compare with other tools/studies and insert the limitations

6.       Check that all the images are with descriptions (adding labels) and that the figures are described in the body of the manuscript.

7.       Check the resolution  of the figures and its actual usefulness

Author Response

(The authors gave the same response as above.)

Round 2

Reviewer 4 Report

N/A